# MEM and MEM4PP: New Tools Supporting the Parallel Generation of Critical Metrics in the Evaluation of Statistical Models

Daniel Homocianu [1],* and Cristina Tîrnăucă [2]

1    Department of Accounting, Business Information Systems, and Statistics, Faculty of Economics and Business Administration, "Alexandru Ioan Cuza" University, 700505 Jassy, Romania
2    Departamento de Matemáticas, Estadística y Computación, Universidad de Cantabria, 39005 Santander, Spain
*    Correspondence: daniel.homocianu@uaic.ro

**Abstract:** This paper describes MEM and MEM4PP as new Stata tools and commands. They support the automatic reporting and selection of the best regression and classification models by adding supplemental performance metrics based on statistical post-estimation and custom computation. In particular, MEM provides helpful metrics, such as the maximum acceptable variance inflation factor (*maxAcceptVIF*) together with the maximum computed variance inflation factor (*maxComputVIF*) for ordinary least squares (OLS) regression, the maximum absolute value of the correlation coefficient in the predictors' correlation matrix (*maxAbsVPMCC*), the area under the curve of receiving operator characteristics (*AUC-ROC*), p and chi-squared of the goodness-of-fit (*GOF*) test for logit and probit, and also the maximum probability thresholds (*maxProbNlogPenultThrsh* and *maxProbNlogLastThrsh*) from Zlotnik and Abraira risk-prediction nomograms (*nomolog*) for logistic regressions. This new tool also performs the automatic identification of the list of variables if run after most regression commands. After simple successive invocations of MEM (in a .do file acting as a batch file), the collectible results are produced in the console or exported to specially designated files (one .csv for all models in a batch). MEM4PP is MEM's version for parallel processing. It starts from the same batch (the same .do file with its path provided as a parameter) and triggers different instances of Stata to parallelly generate the same results (one .csv for each model in a batch). The paper also includes some examples using real-world data from the World Values Survey (the evidence between 1981 and 2020, version number 1.6). They help us understand how MEM and MEM4PP support the testing of predictor independence, reverse causality checks, the best model selection starting from such metrics, and, ultimately, the replication of all these steps.

**Keywords:** regression and classification models; collinearity and reverse causality checks; performance analysis; automation and parallelization tools

## 1. Introduction

In recent years, many concerns regarding the replicability of study findings and data analysis results obtained and reported in scientific publications have emerged. In many cases, one has to re-implement experiments/quasi-experiments to validate the findings and replicate the data analysis or the computation using the same data, procedure, and methodology [1].

Nowadays, many statistical tools such as IBM SPSS Statistics (version 24.0, IBM, Armonk, NY, USA), R Project for Statistical Computing (version 4.2.1, r-project.org, Vienna, Austria), Matlab (version 2022b, The MathWorks Inc., Natick, MA, USA), Minitab (version 19, Minitab, State College, Pennsylvania), SAS (version 9.4, The SAS Institute, Cary, NC, USA), Stata Statistical Software (version 17, Stata Corp, College Station, TX, USA), etc., that enhance replicability by consistently supporting data analysis, statistical calculations, visualizations, and advanced tests and the automatic reporting of results.

MEM is a new command for Stata. It is designed to support the rapid production of tables with many regression and classification models and reduce the time needed to test different variables in different combinations by considering additional performance statistics for each model, such as the maximum absolute value of the correlation coefficients from the correlation matrix of predictors, the largest variance inflation factor, the AUC-ROC [2], and some others related to the goodness-of-fit tests [3].

Moreover, MEM is also designed to rapidly test the predictors' independence to avoid including predictors with the values of the correlation coefficients of 0.7 or more, or that have VIF values of 10 or more, which should correspond to models with a high R-squared (of 0.9 or more, but not less). In such cases, the correlation between the predictors is stronger than the regression relationship, and multicollinearity can affect their coefficient estimates [4]. Furthermore, the MEM command is intended to dispel doubts about possible reverse causality problems through appropriate testing.

The same strengths mentioned above are also valid for MEM4PP (the MEM's version for parallel processing), with the advantage of a reduced reporting time of these critical metrics.

## 2. Related Works

Taking into account that MEM and MEM4PP can be finally used to select variables and statistical models, we can mention here some other useful methods and techniques. For instance, PCA, which stands for principal components analysis [5], allows the estimation of parameters for principal component models. Moreover, it is worth mentioning BMA (Bayesian model averaging) and weighted-average least-squares (*wals*) for estimating linear regression models with uncertainty about the choice of the explanatory variables [6].

As far as Stata is concerned (https://www.stata.com (accessed on 24 August 2022)), the latter has many advantages in terms of aid for statistical analysis, advanced tests, computations, visualizations, and reporting [7–11] and it successfully combines a friendly user interface with support for power users and programmers [12–15].

Moreover, there are many other new Stata programs and commands introduced to serve different purposes. Among them, post estimations such as the AUC-ROC for multinomial regressions (MNL) [16], the calculation of shrinkage statistics to measure overfitting such as *overfit.ado* [17], representations such as risk-prediction nomograms generated using the *nomolog* command [18], the exports of tabulations such as *tabout* [19], and many other specific applications.

In the context of developing MEM and MEM4PP in Stata, it is very important to acknowledge that Stata previously benefited from the *estout* package with support for both the *eststo* and *esttab* commands [20,21], supporting the direct production of tables (in the console and as external files, respectively) with some default performance metrics for well-known statistical models.

When it comes to parallel approaches in general, we can mention early contributions focused on the information gain using MapReduce jobs executed on Hadoop clusters [22], the open-source distributed machine learning library MLib [23], other more recent methods and techniques in Apache Spark [24] and Mahout [25]. In addition, it is worth including other new approaches that focus, in particular, on computing Pearson's correlation coefficients, such as ForkJoinPcc [26].

In terms of parallel approaches in Stata, some previous contributions, carefully tested prior to developing MEM4PP, must be emphasized here: *parallel* [27], *multishell* [28], and *qsub* [29].

## 3. Materials and Methods

Data from World Values Survey (WVS) (The .dta file in WVS TimeSeries between 1981 and 2020, Stata v1 6.zip at: https://www.worldvaluessurvey.org/WVSDocumentationWVL.jsp (accessed on 24 August 2022)) served for proving the usefulness of MEM on real-world data. We started from all variables (1,045) and observations (426,452) in this dataset, which was loaded and exported as .csv using Stata. This export was performed after a simple

binary derivation of the variable to analyze (C033, Job satisfaction) considering the symmetric split of the original scale. Therefore, starting from C033 (original scale of 1 = Dissatisfied .. 10 = Satisfied), we derived C033_bin that contained one for all values greater than or equal to 6 and 0 otherwise (lines 4–6 in the full processing script, Listing A1, Appendix A).

Then, following the steps described in the schematic representation at the end of this paper (Figure A1, Appendix A), the .csv dataset was loaded in the Rattle (https://rattle.togaware.com (accessed on 24 August 2022)) (version 5.4.0) interface of R, and the Adaptive Boosting technique for decision tree classifiers [30] as the 1st round data mining was applied with default settings (Trees:50, Max Depth:6, Min Split:20, Complexity:0.01, Learning Rate:0.3, Threads:2, Iterations:50, and Objective:binary logistic). The purpose of the 1st selection stage was to discover the most important variables related to the one being analyzed (see Figure 1) in its binary form. It was performed on a Windows Server Datacenter virtual machine, configured with a maximum of 32 Intel Xeon Gold 6240 Cascadelake CPU logical cores (from all 36 available–18 physical cores) and 32 gigabytes of RAM, in a private cloud (https://cloud.raas.uaic.ro (accessed on 24 August 2022)) managed using OpenStack on Ubuntu. The reason why the boost plugin in Stata was not used is related to its time-consuming execution and limited capabilities in terms of automatic variable selection and treatment of missing values [31].

**Figure 1.** The results of the 1st round of data mining obtained using the Adaptive Boosting technique in Rattle.

In the second round, we successively invoked two powerful commands in the LASSO package [32] until no loss in selections was observed (the last six non-binary variables are presented in Tables A1 and A2 from Appendix A). Just before the second round, we used the original .dta form of the same WVS dataset and the list of predictors obtained in the first round and created the same binary derivation above (C033_bin, upper side of Listing A1, Appendix A). To perform such consecutive selections, check the original scales, and generate derivations, powerful commands in Stata were used, namely, *rlasso*, which is responsible for controlling overfitting [33], *cvlasso* performs cross-validations on random subsamples [34], *label list*, *generate*, and *replace* (see the processing script, Listing A1, Appendix A). Moreover, aiming for clear and trustful answers and being aware of the traditional treatment procedures for missing values and their effect on classifier accuracy [35], we used elimination conditions for missing values in all derivations ("! =." meaning Not NULL).

Stata 17.0 MP 2021 64-bit was used for all the tests with the MEM tool proposed in this paper. The only additional installations performed were those of the *estout* package



(*ssc install estout, replace*), including support for the use of the *eststo* and *esttab* commands [36], and Zlotnik and Abraira's (2015) [18] nomogram generator, namely, *nomolog* (*net install st0391_1, from* (http://www.stata-journal.com/software/sj15-3) (accessed on 24 August 2022)). The first one was mandatory to benefit from the possibility of automatically generating tables with coefficients and errors corresponding to the regression models by printing them in the console of Stata (approach limited by its space and screen resolution), or even exporting them as .csv files (approach practically unlimited in terms of necessary space and the number of generated rows and columns). The second was also necessary for collecting the last two probability thresholds in the proximity of the model's maximum theoretical probability (maxProbNlogPenultThrsh and maxProbNlogLastThrsh, Figure A2 in Appendix A). Through numerous practical observations based on dragging the perpendiculars and summing up the nomograms' scores [37,38], we found that, in all cases, the maximum theoretical probability was higher than the first limit (maxProbNlogPenultThrsh). Sometimes (the example of single predictor models, top of Figure A2), the maximum probability was even higher than the second threshold (maxProbNlogLastThrsh). Other times (the example of models with more than one predictor, bottom of Figure A2), the maximum theoretical probability was between these two thresholds.

To install MEM is enough to download (the online folder at https://drive.google.com/drive/folders/1d-j6Y1YAMCQTHktMyCrEbIzfrnUe6FhT?usp=sharing (accessed on 24 August 2022)) and copy the mem.ado file (Listing A2, Appendix A) into one of the ado directories (e.g., C:\ado\personal) (https://www.stata.com/manuals13/u17.pdf, section 17.5.2 (accessed on 24 August 2022)). The same applies to MEM4PP (mem4pp.ado, Listing A3, Appendix A) in terms of installation. When designing our own Stata files to automate the generation of tables with classification and regression models and their performance statistics, we relied on the simple logic (Figure 2) of invoking MEM right after a classification or regression command. We have also provided useful examples that can be accessed by following each link to each additional .do file for creating Tables A3–A5 and are available just below each of these three tables (the Appendix A).

**Figure 2.** Simple steps in a script using MEM and meant to produce tables with results (coefficients, errors, and significance) for regression models and also performance metrics such as default statistics together with additional ones brought by MEM.

The first two steps (Figure 2), including the regression command and invocation of the MEM one, must be repeated for each particular model to export. The last one (*esttab*) finally collects (bottom of Figure 2) both default and custom statistics to assess model performance (between the parentheses corresponding to the *stats* keyword). The custom ones (e.g., AUC-ROC) are those computed and/or collected using MEM (see the Listing A2 in Appendix A).

If considered for use in a serial way (single logical processing core involved), a batch (.do file) following the pattern in Figure 2 should be simply invoked with the aid of the do command followed by the path (e.g., do "C:\Table3_rev_cause_checks_logit.do") and only after specifying the dataset.

If considered for use in a parallel way (many logical processing cores involved), a batch following the same pattern above (Figure 2) should be used only as a parameter (the .do file's path) for the second command (MEM4PP based on *qsub* [29], Listing A3, Appendix A). In the second case, there were some keywords considered for exclusion in the design of MEM4PP. For instance, lines starting with the asterisk (*) and indicating comments are ignored. The same for those starting with the invocation of MEM. Additionally, the first line starting with "#*delimit*" causes MEM4PP to ignore it, as well as the commands that come after it (Figure 2, *esttab* and everything up to "#*delimit cr*", which indicates that the carriage return/cr, or enter, was used to split the command line on many rows). Thus, only the remaining lines are executed in parallel (e.g., the lines containing the regression commands for the corresponding models to check, Figure 2, *eststo* model N) and only if they are not distributed on several rows (using the delimitation facility above). Additional parameters for MEM4PP (others except the *dopath* discussed above, Listing A3, lines 3, 6, and 10, Appendix A) are the number of logical processors being allocated (xcpu—2, by default, Listing A3, lines 3, 6, and 23–32, Appendix A) and the specified partition/disk (disk C, by default, Listing A3, lines 3, 6, and 33–42, Appendix A) on which MEM4PP creates a structure of folders used for its specific tasks including the generation of .csv files (Listing A3, lines 43, and 79–87, Appendix A) with model evaluation metrics.

The MEM tool was designed to automatically perform a correlation command and return the maximum absolute value of the predictors' matrix with correlation coefficients (Listing A2, lines 16–32, Appendix A). Therefore, when constructing a script for generating models, the user would not consume extra time and effort to test the predictors' correlation in each model and find this value.

Because of the use of positioning and text extraction functions (Listing A2, lines 7–15, Appendix A) applied to the command line (*e(cmdline)*), MEM automatically isolates the list of predictors and some other specifications next to it, such as the condition to filter the dataset (the *if* clauses in the .do file generating, Table A5, Appendix) by extracting everything between the end of the name of the dependent variable (*e(depvar)*, Listing A2, lines 7, 10, and 13, Appendix A) and the first comma (if any) based on its position (*cpos*). This is because the comma usually serves for introducing additional options for regressions in Stata, such as *vce(robust)* when specifying the computation of robust standard errors to correct for any form of heteroskedasticity [39]. The same reason, related to the position of the 1st comma (Listing A3, lines 71–77, and 118–139, Appendix A), stood behind splitting in two parts (the 2nd and the 3rd corresponding arguments of the dynamically generated task pattern, Listing A3, lines 44–89, Appendix A, namely, "main_do_pp_file.do") each regression or classification command line automatically read by MEM4PP from the batch files (see the .do files indicated via the URLs at the end of the notes of Tables A3–A5, Appendix A). In addition, error checking ("capture ...", "if _rc (_rc stands for the return code built-in variable)==0 ...", "if !missing ...", etc.) was performed after most commands inside MEM and MEM4PP to prevent fatal interruptions in execution and based on the logic of reporting as much as possible.

Therefore, the user's effort is minimized as much as possible by relying on such dynamic extractions performed by MEM.

## 4. Results

The goal of this section is to demonstrate the usefulness of the MEM command in terms of increased support for variable selection with the aid of some tests. These are the tests of reverse causality, collinearity measurements, and comparable performance evaluations of more compact models. Such models have the same number of observations due to applying Not NULL filtering conditions on each variable (from collinear pairs) that is alternatively removed from the more comprehensive but redundant model.

Consequently, starting from the six influences identified at the end of the first two data mining rounds (Figure A1, Appendix A), we performed additional tests considering the same binary form of the outcome (*C033_bin*).

First, we automatically generated a set of 12 single predictor models based on *logit* and MEM (3, Appendix A). We compared, in terms of reverse causality, the models having *C033_bin* as the variable to be analyzed with the inverse ones. The latter had *C033* (job satisfaction on the original scale) as the sole predictor, while the response variable was each time a binary derivation (Listing A1, lines 20–42, Appendix A) corresponding to each of the six influences selected at the end of the first two mining rounds. Only three predictors (*A170*, *C006*, and *C034*) have been confirmed at this point, meaning all six above except *C031* (degree of pride in your work), *C042B1* (why people work: work is like a business transaction), and D002 (satisfaction with home life). According to our results (Table A3, models 5 vs. 6, models 9 vs. 10, and 11 vs. 12), *C031*, *D002*, and *C042B1* are more likely to be considered response variables when analyzed in relation with *C033*. This is because the inverse models indicated better performance: better explanatory power (larger *R-squared*), more information gain and better fit (lower *AIC* and *BIC* values), better accuracy (larger values of *AUC-ROC*), and higher theoretical probability (the same or larger values of *maxProbNlogPenultThrsh* as the penultimate probability threshold; Figure A2, Appendix A).

Next, starting from the three remaining variables, we automatically created an additional set of models based on OLS (ordinary least squares) and MEM (Table A4, Appendix A), with the use of the same variable for analysis (C033_bin). This served to measure the collinearity between each pair of the remaining three predictors (three resulting models) using both the maximum absolute value from the predictors' matrix with correlation coefficients (*maxAbsVPMCC*) and the maximum computed variance inflation factor (*OLSmaxComputVIF*) assessed against the maximum acceptable variance inflation factor (*OLSmaxAcceptVIF*, Equation (1)). Of these three models, only the first (Table A4, model 1) indicated collinearity issues [40,41] between *A170* (life satisfaction) and *C006* (satisfaction with the financial situation of the household). *maxAbsVPMCC* = 0.5643 indicates a moderate correlation between these two predictors. Moreover, *OLSmaxComputVIF* = 1.2797 > *OLSmaxAcceptVIF* = 1.2043. This means that the correlation between the predictors is stronger than the regression relationship [4] and multicollinearity can affect their coefficient estimates [42] (Equation (1)):

$$\text{OLSmaxAcceptVIF} = 1/(1 - \text{Model's R-squared}) \tag{1}$$

Next, we automatically performed additional comparable checks in terms of performance (see Table A5, Appendix A). We conducted this considering more compact models after removing variables that had reverse causality and collinearity issues (Table A3, models 5 vs. 6, 9 vs. 10, and 11 vs. 12, and Table A5, models 1 and 2) previously identified for a pair (Table A4, model 1, and Table A5, models 3 and 4). Moreover, we aimed for a comparable basis regarding the number of valid observations (N), and we introduced a Not NULL filtering condition for each variable being removed from the pair generating collinearity (Table A5, models 5 vs. 6 and 7 vs. 8). That allowed us to objectively compare the resulting models, which performed better when removing *C006* and keeping *A170* (Table A5, models 5, 7, 9, and 11) than vice versa (Table A5, models 6, 8, 10, and 12). The computation of maxProbNlog (both thresholds) this time led to identical results between the two groups of models. The filtering conditions (e.g., if C006! = ./if A1170! = .) were meant to specify the requirement of existing (Not NULL) observations for these variables,

which were alternatively removed from models in order to make the models comparable in terms of support (number of Not NULL intersecting observations, e.g., 15576, Table A5, the pairs of models 3 and 4, 5 and 6, and 7 and 8).

The MEM command was not designed to provide additional performance metrics, when used together with *melogit*, as logistic regression for the multilevel mixed-effects modeling [43]. Therefore, the results of such tests have not been included in this manuscript even if we successfully performed them as additional cross-validations on different but explicit grouping criteria (clusters/random effects). As a result, both remaining predictors after the final step above (*F170* and *C034* from Table A5 as fixed effects) proved to be significant no matter the values of the grouping criteria (respondent's country of origin, age, gender, and marital status).

For all regressions above (Tables A3–A5), we used the *vce(robust)* option, which served for generating robust standard errors and correcting for any form of heteroskedasticity. In addition, MEM successfully succeeded in terms of tests for other types of regressions such as *probit*. For *scobit* (skewed *logit*), *firthlogit*, *Tobit*, *Poisson*, *nbregress* (negative binomial regression), *ologit* (ordinal logistic regression), *oprobit* (ordinal probit regression), and *mlogit* (multinomial logistic regression), MEM returned just the maximum absolute value of the correlation coefficient in predictors' matrices (*maxAbsVPMCC*).

## 5. Discussion

As part of the entire automation approach, MEM could successfully serve to simplify the task of generating tables with regression coefficients and errors, together with the extended performance metrics of the resulting statistical models and, consequently, facilitate the implementation of the scientific principle of reproducibility (repeatability or replicability) or simply providing full support for the replication of results [44] in science and research.

All the results above (Tables A3–A5, Appendix A) can be generated in a fraction of the time consumed by MEM (a single logical processor used), when considering the parallel approach (MEM4PP, see (https://drive.google.com/drive/folders/1FCYiZ_7geagkyN-DAyewdYmD5aSa6l5q?usp=sharing (accessed on 24 August 2022) the three simulations with archives, which contain the resulting .csv files and .pdfs capturing the console) starting from the same batch .do files (those mentioned via their URLs at the end of the notes of Tables A3–A5, Appendix A) specified as the first parameter (*dopath* meaning their local path after download) of MEM4PP. All types of files obtained after performing the simulations above are also meant to provide enough support for replicating the results obtained the same way video captures and tutorials do [45].

In terms of generating the results for a higher number of models (more than 15 in our tests), the Stata console entirely maximized (to the detriment of other areas) on a full-HD screen (1920 × 1080 pixels) failed to generate easily readable results (interlaced values on additional lines) when using the *estout* command. This was the reason why in all our .do script sequences we used *esttab* and automatically produced intelligible results (.csv files). Moreover, when additionally using spreadsheet tools to open the regression models automatically obtained as .csv files, we benefited from useful options to visually format the sets of numerical values when considering rules and corresponding thresholds for the performance metrics above, of which, some are computed and/or collected by MEM. The only condition here was to save the resulting .csv file (one for all models in case of serial/single logical core processing) in the native spreadsheet format (.xlsx), then copy and use the paste special option for values only, and further perform the automatic conversion to the numbers of those cells with the numbers stored as text, which were meant to be visually formatted automatically. In addition, the separate .csv files (one for each model in case of parallel processing) generated by MEM4PP require further integration into a single table. This should be done by the user (the simple addition of empty line pairs and the copy and paste of columns as values), although it usually means extra formatting time required compared to the serial execution scenario (a single resulting .csv file with automatically

and properly aligned columns and lines for all statistical models included when using just MEM, a single logical core for serial processing).

Further projections and ideas related to MEM and MEM4PP consider them a starting point for data mining instruments that are able to automatically explore the best model/models based on any possible combination of predictors in a given list and considering the defined priorities and corresponding weights as parameters. Moreover, they can be improved considering the approaches that are able to compute and report the values of the coefficients [46] indicating the bivariate correlation between the variable to analyze and each independent one [47,48] included in any regression command.

## 6. Conclusions

MEM and its version for parallel processing (MEM4PP) are new tools that bring additional performance metrics to regression and classification models. They rely on both statistical post estimations (e.g., the area under the ROC curve, and p and chi-squared for the goodness-of-fit) and user-defined computations (e.g., the maximum acceptable variance inflation factor versus its maximum calculated/actual value for ordinary least squares regressions, and the maximum absolute value of correlation coefficients in the predictor matrix). These tools also collect the maximum probability thresholds from Zlotnik and Abraira risk prediction nomograms (nomolog) when used after logistic regressions. Moreover, they support extensive automation and parallelization when regression commands couple with the components of the *estout* package, namely, *eststo* and *esttab*. MEM and MEM4PP have also passed many tests for exporting and comparing the dozens and hundreds of models obtained at once (a single .csv file for serial execution) or distinctly (one .csv for each model in the case of parallel processing) by including MEM calls multiple times in the same script (batch example included). This way, both tools exponentially reduce the time required to generate the tables of coefficients and errors for many classification and regression models to report, including the additional performance metrics above. They also facilitate reverse causation checks, collinearity removal, and serve the decision-making process of selecting the best prediction models based on comparative performance criteria. In addition, they both suggest how to overcome the practical limitations related to printing in the main application's console and couple the theoretical and practical advantages of using both statistical and spreadsheet tools. They also open the so-called "Pandora's box" in terms of possibilities for the parallel generation of any customized metric of statistical model performance, including the ones created by applications specialized in generating visual diagnostic representations such as risk-prediction nomograms. Moreover, they even stimulate the implementation of replicability as a scientific principle. A current limitation related only to the second command (MEM4PP) concerns the efficiency of parallel processing in relation to the number of logic cores used simultaneously when overpassing a certain threshold (usually 6 cores), beyond which the parallel loading of the same dataset may generate some delays depending on the storage type and amount, and the performance of both RAM and CPU's cache memory. The necessary optimizations in this direction will be the subject of future related research.

**Author Contributions:** Conceptualization, D.H. and C.T.; methodology, D.H.; software, D.H.; validation, D.H. and C.T.; formal analysis, C.T.; investigation, D.H. and C.T.; resources, D.H.; data curation, D.H.; writing—original draft preparation, D.H.; writing—review and editing, D.H. and C.T.; visualization, D.H.; supervision, C.T.; project administration, D.H.; funding acquisition, D.H. All authors have read and agreed to the published version of the manuscript.

**Funding:** This research did not receive any funding in terms of publishing fees. Still, it benefited from the infrastructure purchased via the projects mentioned in the Acknowledgments section below.

**Institutional Review Board Statement:** The data used in this study belongs to the World Values Survey, which conducted surveys following the Declaration of Helsinki.

**Data Availability Statement:** The dataset used in this study and belonging to the World Values Survey is the .dta file inside the *WVS TimeSeries between 1981 and 2020, Stata v1 6.zip* (https://www.

worldvaluessurvey.org/WVSDocumentationWVL.jsp (accessed on 24 August 2022), the *Data and Documentation* menu, the *Data Download* option, the *Timeseries (1981–2022)* entry) archive.

**Acknowledgments:** For allowing the exploration of the dataset and the agreement to publish the research results, the authors would like to thank the World Values Survey and supporting projects. In terms of technical aid (https://cloud.raas.uaic.ro (accessed on 24 August 2022), as a private cloud of the Alexandru Ioan Cuza University of Iași, Romania), this paper also benefited from the support of the Competitiveness Operational Programme Romania, project number SMIS 124759, RaaS-IS (Research as a Service Iasi) id POC/398/1/124759. This work was also technically supported by the following project we would like to thank: VP50 "Development and validation of software tools and methodologies to provide individualized feedback and automatic performance assessment in programming learning", funded by Consejería de Universidades, Igualdad, Cultura y Deporte del Gobierno de Cantabria.

**Conflicts of Interest:** The authors declare no conflict of interest.

## Appendix A

**Listing A1.** Script with numbered lines (numbers displayed separately, as when opened with the Stata editor) used for performing derivations, executing the 2nd round mining in 3 consecutive LASSO stages (both *rlasso* and *cvlasso*) until no loss in selections, and saving the resulting dataset. (On-line at: https://drive.google.com/u/0/uc?id=1Tmawfhr5py3SvhCyktzP88oWVJvBp1AW&export=download (accessed on 24 August 2022)).

```
1 *Processing script needed to argue the utility of the mem tool in Stata
2 use "F:\WVS_TimeSeries_stata_v1_6.dta"
3 label list C033
4 generate C033_bin = .
5 replace C033_bin = 1 if C033! = . & C033> = 6
6 replace C033_bin = 0 if C033! = . & C033<6 & C033>0
7 ***On the variables selected by Adaptive Boosting (R, the Rattle library) we apply LASSO as
follows:***
8 rlasso C033_bin A170 A173 C006 C031 C034 C042B1 D002 D036 E036 E045 E047
E170_WVS7LOC E180WVS F114A F141 F144 S003 S007 S017 X048WVS X049 Y010 Y020
9 cvlasso C033_bin A170 A173 C006 C031 C034 C042B1 D002 D036 E036 E045 E047
E170_WVS7LOC E180WVS F114A F141 F144 S003 S007 S017 X048WVS X049 Y010 Y020
10 cvlasso, lse
11 ***both rlasso & cvlasso with the lse option (1st LASSO stage above) remove 12 everything
except: A170 A173 C006 C031 C034 C042B1 D002
12 rlasso C033_bin A170 A173 C006 C031 C034 C042B1 D002
13 cvlasso C033_bin A170 A173 C006 C031 C034 C042B1 D002
14 cvlasso, lse
15 ***cvlasso, lse (2nd LASSO stage above) removes A173 after running the previous three
command lines***
16 rlasso C033_bin A170 C006 C031 C034 C042B1 D002
17 cvlasso C033_bin A170 C006 C031 C034 C042B1 D002
18 cvlasso, lse
19 ***cv lasso, lse and rlasso (3rd LASSO stage above) do not eliminate anything (no loss)***
20 label list A170
21 generate A170_bin = .
22 replace A170_bin = 1 if A170! = . & A170> = 6
23 replace A170_bin = 0 if A170! = . & A170<6 & A170>0
24 label list C006
25 generate C006_bin = .
26 replace C006_bin = 1 if C006! = . & C006> = 6
27 replace C006_bin = 0 if C006! = . & C006<6 & C006>0
28 label list C031
29 generate C031_bin = .
30 replace C031_bin = 1 if C031! = . & C031< = 2 & C031>0
31 replace C031_bin = 0 if C031! = . & C031>2
32 label list C034
```

**Listing A1.** *Cont.*

```
33 generate C034_bin = .
34 replace C034_bin = 1 if C034! = . & C034> = 6
35 replace C034_bin = 0 if C034! = . & C034<6 & C034>0
36 *C042B1 is already in a binary form
37 label list C042B1
38 *Therefore, we did not perform any derivation in this case above.
39 label list D002
40 generate D002_bin = .
41 replace D002_bin = 1 if D002! = . & D002> = 6
42 replace D002_bin = 0 if D002! = . & D002<6 & D002>0
43 *Save the resulting dataset (after the processing above)
44 save "F:\WVS_TimeSeries_stata_v1_6processed.dta", replace
45 *Open the resulting dataset above
46 clear all
47 cls
48 use "F:\WVS_TimeSeries_stata_v1_6processed.dta"
```

**Listing A2.** MEM's source script with numbered lines in Stata 17.0 MP 2021 64-bit. (Online at: https://drive.google.com/u/0/uc?id=1t8lGb_mVI2eeWVNqpxFZCTqytnlRQCaz&export=download (accessed on 24 August 2022)).

```
1 *! version 1.1 24August2022
2 *Authors: Daniel HOMOCIANU & Cristina TIRNAUCA
3 *Download "mem.ado" to C:\ado\personal (Run it after any regression by simply using mem)
4 program define mem//(Model Evaluation Metrics)
5 version 17.0
6 ***Section1:Extracting the predictors's list plus other specs.(e.g., condition) upto 1st comma's
position(cpos) from a regression command
7 capture local cpos = strpos(ustrright(e(cmdline),strlen(e(cmdline))-
(strpos(e(cmdline),e(depvar))+strlen(e(depvar)))),",")
8 if _rc = = 0 {
9 if 'cpos' = = 0 {
10 local prdlist_plus =
ustrright(e(cmdline),strlen(e(cmdline))-(strpos(e(cmdline),e(depvar))+strlen(e(depvar))))
11 }
12 if 'cpos'>0 {
13 local prdlist_plus = ustrleft(ustrright(e(cmdline),strlen(e(cmdline))-
(strpos(e(cmdline),e(depvar))+strlen(e(depvar)))),'cpos'-1)
14 }
15 }
16 ***Section2:Computing and storing maxAbsVPMCC = max.Abs.Value from Predictors'Matrix
with Correl.Coefficient
17 ***https://doi.org/10.1213/ane.0000000000002864
18 capture correlate 'prdlist_plus'
19 if _rc = = 0 {
20 matrix crlv = vec(r(C))
21 local lim = rowsof(crlv)
22 local maxAbsVPMCC = 0
23 foreach i of num 1/'lim' {
24 if abs(crlv['i',1])! = 1 {
25 local maxAbsVPMCC = max('maxAbsVPMCC',abs(crlv['i',1]))
26 }
27 }
28 if 'maxAbsVPMCC' = = 0 {
29 local maxAbsVPMCC = .
30 }
31 estadd local maxAbsVPMCC "':di %6.4f 'maxAbsVPMCC'"
32 }
```

**Listing A2.** *Cont.*

```
33 ***Section3:Computing and storing the Variance Inflation Factors (VIFs)—both OLS model's
max.accepted & max.computed VIFs
34 ***https://dx.doi.org/10.4172%2F2161-1165.1000227
35 ***and also AUC(ROC), chi^2 and p for Goodness of Fit (GOF)
36 if e(cmd) = = "regress" {
37 local OLSmaxAcceptVIF = 1/(1-e(r2))
38 estadd local OLSmaxAcceptVIF "':di %6.4f 'OLSmaxAcceptVIF'"
39 capture estat vif
40 if _rc = = 0 {
41 local OLSmaxComputVIF = 'r(vif_1)'
42 estadd local OLSmaxComputVIF "':di %6.4f 'OLSmaxComputVIF'"
43 }
44 }
45 capture lroc, nograph
46 if _rc = = 0 {
47 local AUCROC = 'r(area)'
48 estadd local AUCROC "':di %6.4f 'AUCROC'"
49 }
50 capture estat gof
51 if _rc = = 0 {
52 capture local chi2GOF = 'r(chi2)'
53 if _rc = = 0 {
54 estadd local chi2GOF "':di %6.2f 'chi2GOF'"
55 }
56 capture local pGOF = 'r(p)'
57 if _rc = = 0 {
58 estadd local pGOF "':di %6.4f 'pGOF'"
59 }
60 }
61 ***Section4:Storing last 2 thresholds for maxProbNlog(max.probability on the X axis from
Zlotnik&Abraira's nomogram gen.-nomolog)
62 ***https://doi.org/10.1177%2F1536867 × 1501500212
63 if e(cmd) = = "logit" | e(cmd) = = "logistic" {
64 set graphics off
65 capture nomolog
66 capture local maxProbNlogPenultThrsh = P [1,colsof(P)-1]
67 if _rc = = 0 {
68 estadd local maxProbNlogPenultThrsh "':di %6.4f 'maxProbNlogPenultThrsh'"
69 }
70 capture local maxProbNlogLastThrsh = P [1,colsof(P)]
71 if _rc = = 0 {
72 estadd local maxProbNlogLastThrsh "':di %6.4f 'maxProbNlogLastThrsh'"
73 }
74 set graphics on
75 }
76 end
```

**Listing A3.** MEM4PP's source script with numbered lines in Stata 17.0 MP 2021 64-bit. (Online at: https://drive.google.com/u/0/uc?id=18qjFJvIyhHJI_1AoqO3P3zo4HLglejW0&export=download (accessed on 24 August 2022)).

```
1 *! version 1.1 24August20222
2 *Authors: Daniel HOMOCIANU & Cristina TIRNAUCA
3 *Ex.1: mem4pp, dopath("C:/Table3_rev_cause_checks_logit.do") *Ex.2: mem4pp,
dopath("C:/Table5_comp_perf_checks.do") xcpu(6) disk("C")
4 program define mem4pp
5 version 17.0
6 syntax, dopath(string) [xcpu(real 2) disk(string)]
```

**Listing A3.** *Cont.*

```
7 ***Getting the path of the current dataset and its number of variables***
8 local dataset = "'c(filename)'"
9 local dsetnvars = 'c(k)'+150
10 if 'dsetnvars' < 2048 {
11 local dsetnvars = 2048
12 }
13 if 'dsetnvars' > 120000 {
14 di as error " Error: The dataset is too large (>120000 vars.)!"
15 exit
16 }
17 if missing("'dataset'") {
18 di as error " Error: First you must open a dataset!"
19 exit
20 }
21 ***Checking the existing CPU configuration on the local hardware***
22 local nproc : env NUMBER_OF_PROCESSORS
23 local xc = 2
24 if !missing("'xcpu'") {
25 if 'xcpu'> = 2 & 'xcpu'< = 'nproc' {
26 local xc = int('xcpu')
27 }
28 else {
29 di as error " Error: Provide at least 2 logical CPU cores (but no more than 'nproc') for MP tasks!"
30 exit
31 }
32 }
33 local dsk = "C"
34 if !missing("'disk'") {
35 if "'disk'"< = "z" | "'disk'"< = "Z" {
36 local dsk = "'disk'"
37 }
38 else {
39 di as error "Error: Provide a valid disk letter!"
40 exit
41 }
42 }
43 di "MEM4PP will save the results in the .csv files at 'dsk':\StataPPtasks!"
44 ***Generating the "main_do_pp_file.do" parallel processing template file***
45 local smpt_path = "'dsk':\StataPPtasks\"
46 shell rd "'smpt_path'"/s/q
47 qui mkdir "'smpt_path'"
48 local full_do_path = "'smpt_path'\main_do_pp_file.do"
49 local q_subdir = "queue"
50 qui mkdir '"'smpt_path'/'q_subdir'"'
51 local queue_path = "'smpt_path'\'q_subdir'"
52 local l_subdir = "logs"
53 qui mkdir '"'smpt_path'/'l_subdir'"'
54 local logs_path = "'smpt_path'\'l_subdir'"
55 qui file open mydofile using '"'full_do_path'"', write replace
56 file write mydofile "clear all" _n
57 file write mydofile "log using "
58 file write mydofile '""""'
59 file write mydofile "'logs_path'\log"
60 file write mydofile "'"
61 file write mydofile "1"
62 file write mydofile "'"
63 file write mydofile ".txt"
```

**Listing A3.** *Cont.*

```
64 file write mydofile ‘”””‘
65 file write mydofile “, text” _n
66 file write mydofile “set maxvar ‘dsetnvars’” _n
67 file write mydofile “use “
68 file write mydofile ‘”””‘
69 file write mydofile “‘dataset’”
70 file write mydofile ‘”””‘ _n
71 file write mydofile “”″
72 file write mydofile “2”
73 file write mydofile “”″
74 file write mydofile “, “
75 file write mydofile “”″
76 file write mydofile “3”
77 file write mydofile “”″ _n
78 file write mydofile “mem” _n
79 file write mydofile “esttab model* using “
80 file write mydofile ‘”””‘
81 file write mydofile “‘smpt_path’\model”
82 file write mydofile “”″
83 file write mydofile “1”
84 file write mydofile “”″
85 file write mydofile “.csv”
86 file write mydofile ‘”””‘
87 file write mydofile “, stats(N chi2 p r2 r2_p rmse maxAbsVPMCC OLSmaxAcceptVIF
OLSmaxComputVIF AUCROC pGOF chi2GOF aic bic maxProbNlogPenultThrsh
maxProbNlogLastThrsh) cells(b (star fmt(4)) se(par fmt(4))) starlevels(* 0.05 ** 0.01 *** 0.001)” _n
88 file write mydofile “log close”
89 qui file close mydofile
90 ***Finding the Stata directory***
91 local _sys = “‘c(sysdir_stata)’”
92 local exec : dir “‘_sys’” files “Stata*.exe” , respect
93 foreach exe in ‘exec’ {
94 if inlist(“‘exe’”,”Stata.exe”,”Stata-64.exe”,”StataMP.exe”,”StataMP-
64.exe”,”StataSE.exe”,”StataSE-64.exe”) {
95 local curr_st_exe ‘exe’
96 continue, break
97 }
98 }
99 local st_path = “‘_sys’”+”‘curr_st_exe’”
100 capture confirm file ‘”‘_sys’‘curr_st_exe’”’
101 if _rc ! = 0 {
102 di as error “Stata’s sys dir and executable NOT found!”
103 exit
104 }
105 else {
106 di “!!!Stata’s sys dir and executable found: ‘st_path’ !!!”
107 }
108 ***Dynamically generating and configuring .do files(tasks) using “main_do_pp_file.do”***
109 local k = 0
110 file open myfile using “‘dopath’”, read
111 file read myfile line
112 while r(eof) = = 0 {
113 if “‘ = word(“‘line’”,1)’” = = “#delimit” {
114 continue, break
115 }
116 if “‘ = word(“‘line’”,1)’”! = “mem” & substr(“‘ = word(“‘line’”,1)’”,1,1)! = “*” {
117 local k = ‘k’ + 1
```

**Listing A3.** *Cont.*

```
118 capture local cpos = strpos("'line'",",")
119 if 'cpos' = = 0 {
120 local l_line = "'line'"
121 local r_line = ""
122 }
123 if 'cpos'>0 {
124 local l_line = ustrleft("'line'", 'cpos'-1)
125 local r_line = ustrright("'line'", length("'line'")-'cpos'-1)
126 }
127 if 'k'<10 {
128 qui file open mydofile using 'queue_path'\job0'k'.do, write replace
129 qui file write mydofile '"do "'dsk':\StataPPtasks\main_do_pp_file.do" "0'k'" "'l_line'"
"'r_line'"'
130 }
131 if 'k'> = 10 {
132 qui file open mydofile using 'queue_path'\job'k'.do, write replace
133 qui file write mydofile '"do "'dsk':\StataPPtasks\main_do_pp_file.do" "'k'" "'l_line'"
"'r_line'"'
134 }
135 file close mydofile
136 }
137 file read myfile line
138 }
139 file close myfile
140 ***Allocating .do tasks to logical CPU cores using qsub v.13.1 (06/10/2015), made by Adrian
Sayers.***
141 *ssc install qsub, replace
142 if 'xc'>'k' {
143 local xc = 'k'
144 }
145 qsub , jobdir('queue_path') maxproc('xc') statadir('st_path') deletelogs
146 ***Printing logs for all .do tasks in the main session's console***
147 local mylogs : dir "'logs_path'" files "*.txt"
148 local k = 0
149 foreach entry in 'mylogs' {
150 local k = 'k' + 1
151 if 'k'<10 {
152 type "'logs_path'\log0'k'.txt"
153 }
154 if 'k'> = 10 {
155 type "'logs_path'\log'k'.txt"
156 }
157 }
158 end
```

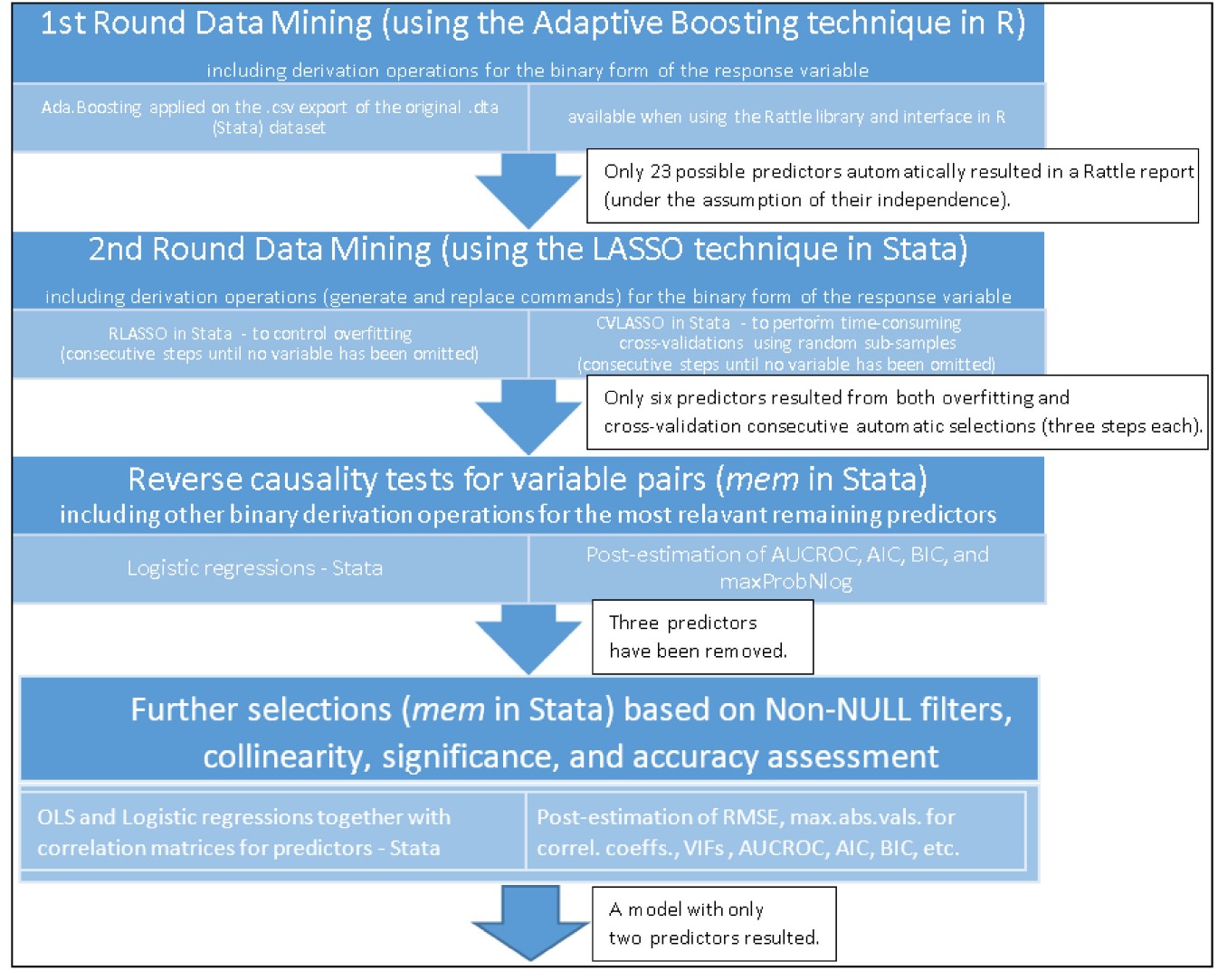

**Figure A1.** Schematic illustration of the techniques used. Source: The authors' projection.

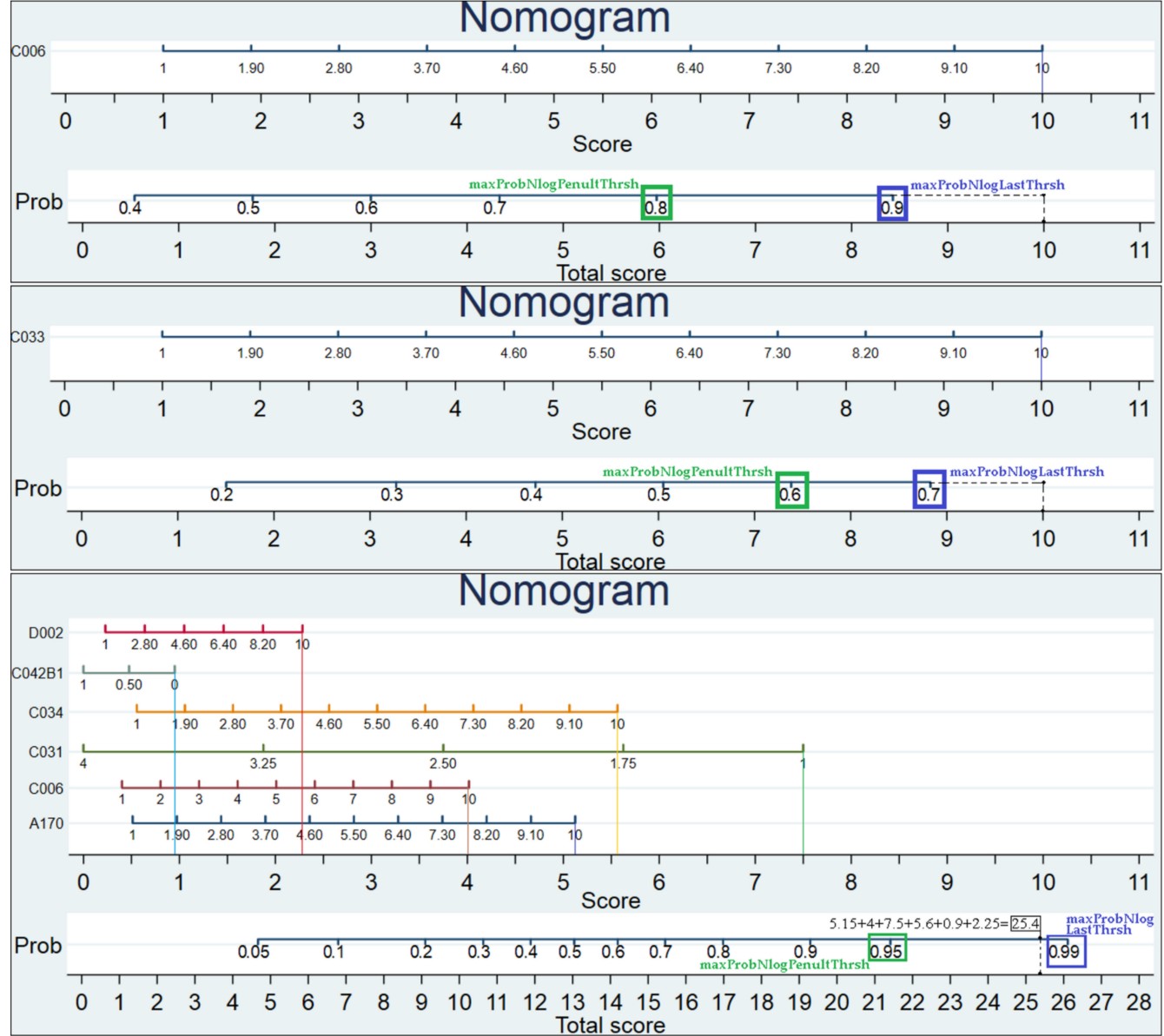

**Figure A2.** Illustration of the last two probability thresholds in the proximity of the model's maximum theoretical probability using three nomograms for Table A3 (models 3 and 4) and Table A5 (model 1). Source: Own calculations using the nomolog command in Stata 17.0 MP 2021 64-bit.

**Table A1.** Important World Values Survey items and corresponding variables, namely, the dependent one (two forms) and six independently selected at the end of the first two mining rounds (Figure A1, Appendix A).

| Variable | Question | Coding |
|---|---|---|
| C033 | Job satisfaction—DEPENDENT VARIABLE | 1-Dissatisfied . . . 10-Satisfied |
| C033_bin | Job satisfaction (binary format)—DEPENDENT VARIABLE | 1 if C033! = . & C033> = 6<br>0 if C033! = . & C033<6 & C033>0 |
| A170 | Satisfaction with your life | 1-Dissatisfied . . . 10-Satisfied |
| A170_bin | Satisfaction with your life (binary format) | 1 if A170! = . & A170> = 6<br>0 if A170! = . & A170<6 & A170>0 |
| C006 | Satisfaction with the financial situation of household | 1-Dissatisfied . . . 10-Satisfied |
| C006_bin | Satisfaction with the financial situation of household (binary format) | 1 if C006! = . & C006> = 6<br>0 if C006! = . & C006<6 & C006>0 |
| C031 | Degree of pride in your work | 1-A great deal . . . 4-None |
| C031_bin | Degree of pride in your work (binary format) | 1 if C031! = . & C031< = 2 & C031>0<br>0 if C031! = . & C031>2 |
| C034 | Freedom of decision-taking in the job | 1-Not at all . . . 10-A great deal |
| C034_bin | Freedom of decision-taking in the job (binary format) | 1 if C034! = . & C034> = 6<br>0 if C034! = . & C034<6 & C034>0 |
| C042B1 | Why people work: work is like a business transaction | 0-Not mentioned, 1-Mentioned |
| D002 | Satisfaction with home life | 1-Dissatisfied . . . 10-Satisfied |
| D002_bin | Satisfaction with home life (binary format) | 1 if D002! = . & D002> = 6<br>0 if D002! = . & D002<6 & D002>0 |

Source: World Values Survey and own calculations in Stata 17.0 MP 2021 64-bit using the following commands: *label list*, *generate*, and *replace*.

**Table A2.** Descriptive statistics for the World Values Survey's variables, namely, the dependent one (two forms) and six independently selected at the end of the first two mining rounds (Figure A1, Appendix A).

| Variable | N | Mean | Std.Dev. | Min | 0.25 | Median | 0.75 | Max |
|---|---|---|---|---|---|---|---|---|
| C033 | 15,968 | 7.27 | 2.31 | 1 | 6 | 8 | 9 | 10 |
| C033_bin | 15,968 | 0.77 | 0.42 | 0 | 1 | 1 | 1 | 1 |
| A170 | 420,669 | 6.7 | 2.42 | 1 | 5 | 7 | 8 | 10 |
| A170_bin | 420,669 | 0.69 | 0.46 | 0 | 0 | 1 | 1 | 1 |
| C006 | 411,461 | 5.75 | 2.58 | 1 | 4 | 6 | 8 | 10 |
| C006_bin | 411,461 | 0.54 | 0.5 | 0 | 0 | 1 | 1 | 1 |
| C031 | 14,988 | 1.73 | 0.87 | 1 | 1 | 2 | 2 | 4 |
| C031_bin | 14,988 | 0.51 | 0.5 | 0 | 0 | 1 | 1 | 1 |
| C034 | 17,900 | 6.54 | 2.79 | 1 | 5 | 7 | 9 | 10 |
| C034_bin | 17,900 | 0.65 | 0.48 | 0 | 0 | 1 | 1 | 1 |
| C042B1 | 22,493 | 0.14 | 0.35 | 0 | 0 | 0 | 0 | 1 |
| D002 | 25,653 | 7.72 | 2.24 | 1 | 7 | 8 | 10 | 10 |
| D002_bin | 25,653 | 0.83 | 0.38 | 0 | 1 | 1 | 1 | 1 |

Source: Own calculations in Stata 17.0 MP 2021 64-bit using the *univar* command.

**Table A3.** Comparative reverse causality checks using logistic regressions for job satisfaction and each potential predictor from the 6 resulting from the 2nd round mining.

| Model Predictors/Response Var. | (1) C033_bin | (2) A170_bin | (3) C033_bin | (4) C006_bin | (5) C033_bin | (6) C031_bin | (7) C033_bin | (8) C034_bin | (9) C033_bin | (10) C042B1 | (11) C033_bin | (12) D002_bin |
|---|---|---|---|---|---|---|---|---|---|---|---|---|
| **A170** | 0.3973 *** −0.0097 | | | | | | | | | | | |
| **C006** | | | 0.3300 *** −0.0084 | | | | | | | | | |
| **C031** | | | | | −1.2461 *** −0.0263 | | | | | | | |
| **C034** | | | | | | | 0.3233 *** −0.0077 | | | | | |
| **C042B1** | | | | | | | | | −0.7301 *** −0.0508 | | | |
| **D002** | | | | | | | | | | | 0.3264 *** −0.0092 | |
| **C033** | | 0.3480 *** −0.0089 | | 0.3049 *** −0.0081 | | 0.5306 *** −0.0118 | | 0.3800 *** −0.0088 | | −0.1425 *** −0.0099 | | 0.3360 *** −0.0096 |
| **_cons** | −1.3840 *** −0.0646 | −1.2868 *** −0.0618 | −0.5840 *** −0.0473 | −1.8448 *** −0.0604 | 3.5825 *** −0.0576 | −1.8024 *** −0.0729 | −0.7322 *** −0.0475 | −1.9542 *** −0.0638 | 1.3594 *** −0.0233 | −0.6646 *** −0.0705 | −1.1913 *** −0.069 | −0.6141 *** −0.0644 |
| N | 15,848 | 15,848 | 15,811 | 15,811 | 14,900 | 14,900 | 15,811 | 15,811 | 13,528 | 13,528 | 15,752 | 15,752 |
| chi^2 | 1681.697 | 1511.46 | 1558.748 | 1406.843 | 2237.2495 | 2034.758 | 1771.526 | 1851.932 | 206.4193 | 208.5788 | 1253.118 | 1212.24 |
| p | 0 | 0 | 0 | 0 | 0 | 0 | 0 | 0 | 0 | 0 | 0 | 0 |
| pseudo R^2 | 0.1258 | 0.106 | 0.1046 | 0.08 | 0.1833 | 0.2168 | 0.1244 | 0.1204 | 0.0135 | 0.0178 | 0.088 | 0.0993 |
| **AUC-ROC** | 0.7443 | 0.7129 | 0.7272 | 0.6797 | 0.7667 | 0.8095 | 0.7377 | 0.728 | 0.5548 | 0.5927 | 0.6912 | 0.7193 |
| AIC | 14,832.35 | 15,908.21 | 15,176.62 | 19,733.43 | 13,249.447 | 10,656.06 | 14,786.66 | 17,607.53 | 14,307.509 | 11,673.32 | 15,391.21 | 12,641.82 |
| BIC | 14,847.69 | 15,923.55 | 15,191.96 | 19,748.77 | 13,264.665 | 10,671.28 | 14,802 | 17,622.87 | 14,322.534 | 11,688.34 | 15,406.54 | 12,657.15 |
| **maxProbNlogPenultThrsh** | 0.8 | 0.7 | 0.8 | 0.6 | 0.9 | 0.9 | 0.8 | 0.7 | ···. | 0.2 | 0.7 | 0.8 |
| **maxProbNlogLastThrsh** | 0.9 | 0.8 | 0.9 | 0.7 | 0.95 | 0.95 | 0.9 | 0.8 | 0.7 | 0.3 | 0.8 | 0.9 |

Source: Own calculations in Stata 17.0 MP 2021 64-bit. Notes: Robust standard errors are presented in parentheses. All raw coefficients above parentheses emphasized using *** are significant at 1‰. The additional performance statistics automatically obtained using MEM are emphasized (bold). Green vs. red indicate better comparative performance and, consequently, variables more likely to be predictors or response variables. The .do script used for generating this table is available online at: https://drive.google.com/u/0/uc?id=1mR2X-psHG4IhUlTlfQL9s7Y5iEMQHygY&export=download (accessed on 24 August 2022).

**Table A4.** Collinearity checks using ordinary least squares regressions for job satisfaction and each pair of the 3 remaining predictors (1st column, Table A3, Appendix A).

| Model | (1) | (2) | (3) |
|---|---|---|---|
| **Predictors/Response Var.** | C033_bin | | |
| A170 | 0.0512 *** | 0.0536 *** | |
| | −0.0017 | −0.0015 | |
| C006 | 0.0338 *** | | 0.0409 *** |
| | −0.0014 | | −0.0013 |
| C034 | | 0.0433 *** | 0.0451 *** |
| | | −0.0013 | −0.0013 |
| _cons | 0.2089 *** | 0.1061 *** | 0.2267 *** |
| | −0.0123 | −0.0125 | −0.0111 |
| N | 15704 | 15705 | 15671 |
| p | 0 | 0 | 0 |
| R^2 | 0.1697 | 0.2111 | 0.1906 |
| RMSE | 0.3814 | 0.371 | 0.3759 |
| **maxAbsVPMCC** | **0.5643** | 0.2765 | 0.2759 |
| **OLSmaxAcceptVIF** | 1.2043 | 1.2676 | 1.2355 |
| **OLSmaxComputVIF** | **1.2797** | 1.0873 | 1.0882 |
| AIC | 14,298.3711 | 13,423.0338 | 13,806.915 |
| BIC | 14,321.3561 | 13,446.019 | 13,829.8937 |

Source: Own calculations in Stata 17.0 MP 2021 64-bit. Notes: Robust standard errors are presented in parentheses. All raw coefficients above parentheses emphasized using *** are significant at 1‰. The additional performance statistics automatically obtained using MEM are emphasized (bold). Red indicates unacceptable collinearity (OLSmaxComputVIF > OLSmaxAcceptVIF) or moderate correlation between predictors (maxAbsVPMCC). The .do script used for generating this table is available online at: https://drive.google.com/u/0/uc?id=18bFCkH4 8BbXFe-KMzxiQAT3ehoEIH09A&export=download (accessed on 24 August 2022).

Table A5. Comparative models for predicting job satisfaction (C033_bin) after removing reverse causality and collinearity issues and performing additional checks.

| Model | (1) | (2) | (3) | (4) | (5) | (6) | (7) | (8) | (9) | (10) | (11) | (12) |
|---|---|---|---|---|---|---|---|---|---|---|---|---|
| Regression Type | logit | OLS | logit | OLS | logit | logit | OLS | OLS | logit | logit | OLS | OLS |
| Filter Condition | N/A | N/A | N/A | N/A | if C006! = . | if A170! = . | if C006! = . | if A170! = . | N/A | N/A | N/A | N/A |
| Predictors/Response Var. | C033_bin | | | | | | | | | | | |
| A170 | 0.1802 *** | 0.0244 *** | 0.2667 *** | 0.0416 *** | 0.3433 *** | | 0.0535 *** | | 0.3441 *** | | 0.0536 *** | |
| | −0.0141 | −0.0019 | −0.0111 | −0.0017 | −0.0103 | | −0.0016 | | −0.0102 | | −0.0015 | |
| C006 | 0.1413 *** | 0.0166 *** | 0.1851 *** | 0.0254 *** | | 0.2780 *** | | 0.0409 *** | | 0.2776 *** | | 0.0409 *** |
| | −0.0122 | −0.0015 | −0.0102 | −0.0014 | | −0.0092 | | −0.0013 | | −0.0091 | | −0.0013 |
| C031 | −0.8791 *** | −0.1425 *** | | | | | | | | | | |
| | −0.0305 | −0.0047 | | | | | | | | | | |
| C034 | 0.1957 *** | 0.0261 *** | 0.2579 *** | 0.0394 *** | 0.2784 *** | 0.2765 *** | 0.0432 *** | 0.0449 *** | 0.2791 *** | 0.2768 *** | 0.0433 *** | 0.0451 *** |
| | −0.0099 | −0.0014 | −0.0084 | −0.0013 | −0.0083 | −0.0081 | −0.0013 | −0.0013 | −0.0083 | −0.0081 | −0.0013 | −0.0013 |
| C042B1 | −0.3345 *** | −0.0523 *** | | | | | | | | | | |
| | −0.0653 | −0.0092 | | | | | | | | | | |
| D002 | 0.0803 *** | 0.0123 *** | | | | | | | | | | |
| | −0.0136 | −0.002 | | | | | | | | | | |
| _cons | −0.7354 *** | 0.4882 *** | −3.1023 *** | 0.0649 *** | −2.7134 *** | −1.9714 *** | 0.1076 *** | 0.2276 *** | −2.7222 *** | −1.9723 *** | 0.1061 *** | 0.2267 *** |
| | −0.1383 | −0.0219 | −0.0866 | −0.0127 | −0.0813 | −0.0672 | −0.0126 | −0.0112 | −0.081 | −0.0669 | −0.0125 | −0.0111 |
| N | 12,899 | 12,899 | 15,576 | 15,576 | 15,576 | 15,576 | 15,576 | 15,576 | 15,705 | 15,671 | 15,705 | 15,671 |
| chi2 | 2477.4685 | | 2541.0448 | | 2376.3159 | 2285.7133 | | | 2400.0313 | 2306.7919 | | |
| p | 0 | | 0 | 0 | 0 | 0 | 0 | 0 | 0 | 0 | 0 | 0 |
| R^2 | | 0.2997 | | 0.2279 | | | 0.2101 | 0.19 | | | 0.2111 | 0.1906 |
| pseudo R^2 | 0.2927 | | 0.2231 | | 0.2021 | 0.1842 | | | 0.203 | 0.1846 | | |
| RMSE | | 0.3487 | | 0.3668 | | | 0.371 | 0.3757 | | | 0.371 | 0.3759 |
| **maxAbsVPMCC** | 0.5264 | 0.5264 | 0.4696 | 0.4696 | 0.2763 | 0.2754 | 0.2763 | 0.2754 | 0.2765 | 0.2759 | 0.2765 | 0.2759 |
| **OLSmaxAcceptVIF** | | 1.4279 | | 1.2951 | | | 1.2659 | 1.2346 | | | 1.2676 | 1.2355 |
| **OLSmaxComputVIF** | | 1.5845 | | 1.3203 | | | 1.0872 | 1.0878 | | | 1.0873 | 1.0882 |
| **AUC−ROC** | 0.852 | | 0.8166 | | 0.8022 | 0.7919 | | | 0.8028 | 0.7922 | | |
| **p GOF** | 0.0017 | | 0 | | 0 | 0 | | | 0 | 0 | | |
| **chi^2 GOF** | 7159.82 | | 1155.72 | | 283.98 | 256.74 | | | 280.01 | 260.9 | | |
| AIC | 9709.2467 | 9434.8801 | 12,902.2457 | 12,963.9656 | 13,249.4442 | 13,546.3796 | 13,317.1563 | 13,707.6119 | 13,353.616 | 13,640.0974 | 13,423.0338 | 13,806.915 |
| BIC | 9761.501 | 9487.1345 | 12,932.8596 | 12,994.5796 | 13,272.4047 | 13,569.3401 | 13,340.1168 | 13,730.5724 | 13,376.6012 | 13,663.0761 | 13,446.019 | 13,829.8937 |

**Table A5.** *Cont.*

| Model | (1) | (2) | (3) | (4) | (5) | (6) | (7) | (8) | (9) | (10) | (11) | (12) |
|---|---|---|---|---|---|---|---|---|---|---|---|---|
| **Regression Type** | logit | OLS | logit | OLS | logit | logit | OLS | OLS | logit | logit | OLS | OLS |
| **Filter Condition** | N/A | N/A | N/A | N/A | <span style="color:green">if C006! = .</span> | <span style="color:red">if A170! = .</span> | <span style="color:green">if C006! = .</span> | <span style="color:red">if A170! = .</span> | N/A | N/A | N/A | N/A |
| **Predictors/Response Var.** | C033_bin | | | | | | | | | | | |
| **maxProbNlogPenultThrsh** | 0.95 | | 0.9 | | 0.9 | 0.9 | | | 0.9 | 0.9 | | |
| **maxProbNlogLastThrsh** | 0.99 | | 0.95 | | 0.95 | 0.95 | | | 0.95 | 0.95 | | |

Source: Own calculations in Stata 17.0 MP 2021 64-bit. Notes: Robust standard errors are presented in parentheses. All raw coefficients above parentheses emphasized using *** are significant at 1‰. The additional performance statistics automatically obtained using MEM are emphasized (bold). Green vs. red indicates better comparative performance and, consequently, better models. Red alone indicates unacceptable collinearity (OLSmaxComputVIF > OLSmaxAcceptVIF) or moderate correlation between predictors (maxAbsVPMCC). The .do script used for generating this table is available online at: https://drive.google.com/u/0/uc?id=1lsqxomXQX8mVlbtemRAnF9OHProX1Wfn&export=download (accessed on 24 August 2022).

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
