# Peer review of "MEM and MEM4PP: New Tools Supporting the Parallel Generation of Critical Metrics in the Evaluation of Statistical Models"

_axioms, doi:10.3390/axioms11100549_

Round 1
Reviewer 1 Report
This article presents MEM and its version for parallel processing (MEM4PP) are new tools that bring additional performance metrics to regression and classification models. Said tools also collect the maximum probability limits of Zlotnik and Abraira
risk prediction nomograms (nomolog) when used after logistic regressions.
They support automation and extensive parallelization when regression commands are coupled to the components of the estout package, i.e. eststo si esttab. MEM and MEM4PP also pass many tests to export and compare tens and hundreds of models taken at once (a single .csv file for serial execution) or separately (one .csv for each
model in case of parallel processing), including MEM calls multiple times in the same
script (batch example included). Both tools exponentially reduce the time needed to generate the tables of coefficients and errors to report for many classification and regression models, including the additional performance metrics above more, they facilitate reverse cause checks, remove collinearity and serve the decision making process of selecting the best forecasting models based on comparative performance criteria. The article is well written and well formulated, with new contributions; therefore the publication is accepted.
Author Response
Responses to Reviewer No.1
October 2022
Dear Reviewer,
Thank you very much for accepting to review this manuscript id axioms-1918342 entitled “MEM and MEM4PP: New tools supporting the parallel generation of critical metrics in the evaluation of statistical models” (authors - Daniel HOMOCIANU and Cristina TÎRNÄ‚UCÄ‚), sent for consideration for publication in Axioms (the first round of revisions).
Regarding your questions/suggestions/appreciations, we will address each as follows:
Reviewer 1:
Are the results clearly presented? Can be improved (x)
Authors’ response:
Thank you very much for your advice. The manuscript has been revised again in terms of language and style using the Grammarly well-known tool. All correctness / critical alerts have been solved. Moreover, the introduction of a section dedicated to Related Works (15 extra references included), improved resolution for Figure 2, the split between Results and Discussion together with some changes and additions in the main text (track changes enabled and green background for all revisions) will contribute to improving the presentation.
Reviewer 1:
This article presents MEM and its version for parallel processing (MEM4PP) are new tools that bring additional performance metrics to regression and classification models. Said tools also collect the maximum probability limits of Zlotnik and Abraira risk prediction nomograms (nomolog) when used after logistic regressions.
They support automation and extensive parallelization when regression commands are coupled to the components of the estout package, i.e., eststo si esttab. MEM andMEM4PP also pass many tests to export and compare tens and hundreds of models taken at once (a single .csv file for serial execution) or separately (one .csv for each model in case of parallel processing), including MEM calls multiple times in the same script (batch example included). Both tools exponentially reduce the time needed to generate the tables of coefficients and errors to report for many classification and regression models, including the additional performance metrics above, they facilitate reverse cause checks, remove collinearity and serve the decision-making process ofselecting the best forecasting models based on comparative performance criteria.
The article is well written and well formulated, with new contributions; therefore, the publication is accepted.
Authors’ response:
Thank you very much for your kind words of appreciation.
Thank you for the time and consideration given to this submission and for helping us to improve it. We look forward to your reply.
Sincerely,
Cristina TÎRNÄ‚UCÄ‚, Profesora Contratado Doctor,
Departamento de Matemáticas, Estadística y Computación,
Facultad de Ciencias, Av. de los Castros s/n, Universidad de Cantabria,
Tel. +34 942 203 941, cristina.tirnauca@unican.es
Daniel HOMOCIANU, Associate Professor, Ph.D.,
Alexandru Ioan Cuza University, Iasi, Romania
Faculty of Economics and Business Administration,
Department of Accounting, Business Information Systems, and Statistics,
daniel.homocianu@uaic.ro daniel.homocianu@feaa.uaic.ro

Reviewer 2 Report
My comments:
1. The topic of this paper is interesting and it will contribute in related research field.
2. A section of “Related Works” or “Literature Review” is necessary for this paper.
3. I suggest to separate “Result” and “Discussion” into two sections. The authors have to strengthen them even more, because they are the core of a paper.
4. The section “Conclusions” must be reinforced more. For example, the more contributions to academic research as well as theoretical implications, research limitations, and suggestions for further research.
Author Response
Responses to Reviewer No.2
October 2022
Dear Reviewer,
Thank you very much for accepting to review this manuscript id axioms-1918342 entitled “MEM and MEM4PP: New tools supporting the parallel generation of critical metrics in the evaluation of statistical models” (authors - Daniel HOMOCIANU and Cristina TÎRNÄ‚UCÄ‚), sent for consideration for publication in Axioms (the first round of revisions).
Regarding your questions/suggestions/appreciations, we will address each as follows:
Reviewer 2:
- The topic of this paper is interesting, and it will contribute to the related research field.
Authors’ response:
Thank you very much for your kind words of appreciation.
Reviewer 2:
- A section of “Related Works” or “Literature Review” is necessary for this paper.
Authors’ response:
Thank you very much for your observation. We performed the addition of this section (Related Works) as suggested with 15 extra references included. The changes and additions in the main text considered the track changes option (enabled) and green background for all revisions.
Reviewer 2:
- I suggest separating “Result” and “Discussion” into two sections. The authors have tostrengthen them even more because they are the core of a paper.
Authors’ response:
Thank you very much for your observations. We performed the split as suggested. Moreover, we improved both sections as indicated (track changes option enabled and green background for all revisions).
Reviewer 2:
- The section “Conclusions” must be reinforced more. For example, the more contributions to academic research as well as theoretical implications, research limitations, and suggestions for further research.
Authors’ response:
Thank you very much for your observations. As suggested, we improved the conclusions section in the revised version of the manuscript aiming for theoretical implications, research limitations, and further research (track changes option enabled and green background for all revisions).
Thank you for the time and consideration given to this submission and for helping us to improve it. We look forward to your reply.
Sincerely,
Cristina TÎRNÄ‚UCÄ‚, Profesora Contratado Doctor,
Departamento de Matemáticas, Estadística y Computación,
Facultad de Ciencias, Av. de los Castros s/n, Universidad de Cantabria,
Tel. +34 942 203 941, cristina.tirnauca@unican.es
Daniel HOMOCIANU, Associate Professor, Ph.D.,
Alexandru Ioan Cuza University, Iasi, Romania
Faculty of Economics and Business Administration,
Department of Accounting, Business Information Systems, and Statistics,
daniel.homocianu@uaic.ro daniel.homocianu@feaa.uaic.ro

Reviewer 3 Report
The majority of the publication is in the Appendix - it seems that this is more of a "Technical Paper" that must be presented in a conference proceeding after a Demo than a journal. What's the scientific contribution? It is certainly a scientific tool many may use but there is nothing specific to this tool that did not exist before!
Author Response
Responses to Reviewer No.3
October 2022
Dear Reviewer,
Thank you very much for accepting to review this manuscript id axioms-1918342 entitled “MEM and MEM4PP: New tools supporting the parallel generation of critical metrics in the evaluation of statistical models” (authors - Daniel HOMOCIANU and Cristina TÎRNÄ‚UCÄ‚), sent for consideration for publication in Axioms (the first round of revisions).
Regarding your questions/suggestions/appreciations, we will address each as follows:
Reviewer 3:
The majority of the publication is in the Appendix - it seems that this is more of a"Technical Paper" that must be presented in a conference proceeding after a Demo than ajournal. What's the scientific contribution? It is certainly a scientific tool many may use but there is nothing specific to this tool that did not exist before!
Authors’ response:
Thank you very much for your observations.
The methodology and the tools described in this paper were also confirmed when presenting previous papers containing results validated using these tools.
For instance:
-Reverse causality issues. The case of financial satisfaction
- life satisfaction. Global evidence from the World Values Survey, presented at GEBA 2021
(Chairman: prof. Marin Fotache). An incipient form of the paper is available online at: https://dx.doi.org/10.2139/ssrn.4005824
-Jobs Scarce - Predictors for the Nation's Priority over Migrants.
Evidence from World Values Survey, presented at
AMEC CONFERENCE 2021, Track: Data Analysis in the Industry 4.0 Era
(Chairman: prof. Pawel Lula) https://amec.hse.ru/mirror/pubs/share/535424152
-Determinants of job satisfaction with respect to dominant residence and education before the end of the Golden Era. Evidence from SHARE-ERIC (Wave 7), presented at
AMEC CONFERENCE 2020, Track: Data Analysis in the Industry 4.0 Era
(Chairman: prof. Pawel Lula) https://amec.hse.ru/mirror/pubs/share/411254986
The latter was developed and published in a scientific journal: https://rdcu.be/cnXvc
Regarding the questions and suggestions, we performed the addition of a dedicated section (Related Works - 15 extra references included) to better emphasize the existing contributions.
More, we performed some updates in the section dedicated to Materials and Methods. We also improved the resolution of Figure 2.
More, we performed a split between Results and Discussion and improved the resulting two in order to better emphasize the advantages of both tools.
We also improved the conclusions section in the revised version of the manuscript aiming for theoretical implications, research limitations, and further contributions.
All changes and additions performed in the revised version of the manuscript considered the track changes option (enabled) and green background for all revisions.
Thank you for the time and consideration given to this submission and for helping us to improve it. We look forward to your reply.
Sincerely,
Cristina TÎRNÄ‚UCÄ‚, Profesora Contratado Doctor, Departamento de Matemáticas, Estadística y Computación, Facultad de Ciencias, Av. de los Castros s/n, Universidad de Cantabria, Tel. +34 942 203 941, cristina.tirnauca@unican.es
Daniel HOMOCIANU, Associate Professor, Ph.D., Alexandru Ioan Cuza University, Iasi, Romania, Faculty of Economics and Business Administration, Department of Accounting, Business Information Systems, and Statistics, daniel.homocianu@uaic.ro

Round 2
Reviewer 3 Report
N/A